# On Alternative Algorithms for Computing Dynamic Mode Decomposition

Gyurhan Nedzhibov

Faculty of Mathematics and Informatics, Shumen University, 9700 Shumen, Bulgaria; g.nedzhibov@shu.bg

**Abstract:** Dynamic mode decomposition (DMD) is a data-driven, modal decomposition technique that describes spatiotemporal features of high-dimensional dynamic data. The method is equation-free in the sense that it does not require knowledge of the underlying governing equations. The main purpose of this article is to introduce new alternatives to the currently accepted algorithm for calculating the dynamic mode decomposition. We present two new algorithms which are more economical from a computational point of view, which is an advantage when working with large data. With a few illustrative examples, we demonstrate the applicability of the introduced algorithms.

**Keywords:** dynamic mode decomposition; Koopman operator; singular value decomposition; equation-free; Frobenius companion matrix

## 1. Introduction

Dynamic mode decomposition (DMD) was first introduced by Schmid [1] as a method for analyzing data from numerical simulations and laboratory experiments in fluid dynamics field. The method constitutes a mathematical technique for identifying spatiotemporal coherent structures from high-dimensional data. It can be considered to be a numerical approximation to Koopman spectral analysis, and in this sense, it is applicable to nonlinear dynamical systems (see [2–4]). The DMD method combines the favorable features from two of the most powerful data analytic tools: proper orthogonal decomposition (POD) in space and Fourier transforms in time. DMD has gained popularity and it has been applied for a variety of dynamic systems in many different fields such as video processing [5], epidemiology [6], neuroscience [7], financial trading [8–10], robotics [11], cavity flows [12,13] and various jets [2,14]. For a review of the DMD literature, we refer the reader to [15–18]. Since its initial introduction, along with its wide application in various fields, the DMD method has undergone various modifications and improvements. For some recent results on the topics of DMD for non-uniformly sampled data, higher order DMD method, parallel implementations of DMD and some derivative DMD techniques, we recommend to the reader [19–26]; see also [27–33].

Our goal in the present work is to introduce alternative algorithms for calculating DMD. The new approaches calculate the DMD modes of the Koopman operator using a simpler formula compared to the standard DMD algorithm. The remainder of this work is organized as follows: in the rest of Section 1, we briefly describe the DMD algorithm, in Section 1, we propose and discuss the new approaches for DMD computation and in Section 3, we present numerical results, and the conclusion is in Section 4.

### 1.1. Description of the Standard DMD Algorithm

Originally, the DMD technique was formulated in terms of a companion matrix [1,2], emphasizing its connections to the Arnoldi algorithm and Koopman operator theory. Later, an SVD-based algorithm was presented in [12]. This algorithm is more numerically stable and is now a commonly accepted approach for performing DMD decomposition. We describe this algorithm in the following. Throughout the paper, we use the following notations: uppercase Latin letters for matrices, lowercase Latin or Greek letters for scalars, and lowercase bold letters for vectors.

Consider a sequential set of data arranged in $n \times m + 1$ matrix

$$Z = [\mathbf{x}_0, \ldots, \mathbf{x}_m] \tag{1}$$

where $n$ is the number of state variables and $m + 1$ is the number of observations (*snapshots*). The data $\mathbf{x}_i$ could be from measurements, experiments, or simulations collected at the time $t_i$ from a given dynamical system and assume that the data are equispaced in time, with a time, step $\triangle t$. We assume that the data $Z$ are generated by linear dynamics, i.e., assume that there exists a linear operator $A$ such that

$$\mathbf{x}_{k+1} = A\mathbf{x}_k, \quad \text{for } k = 0, \ldots, m - 1. \tag{2}$$

The goal of the DMD method is to find an eigendecomposition of the (unknown) operator $A$. To proceed, we use an arrangement of the data set into two large data matrices

$$X = [\mathbf{x}_0, \ldots, \mathbf{x}_{m-1}] \quad \text{and} \quad Y = [\mathbf{x}_1, \ldots, \mathbf{x}_m]. \tag{3}$$

Therefore, the xpression (2) is equivalent to

$$Y = AX. \tag{4}$$

Then the DMD of the data matrix $Z$ is given by the eigendecomposition of $A$. We can approximate operator $A$ by using singular value decomposition (SVD) of data matrix $X = U\Sigma V^*$, where $U$ is an $n \times n$ unitary matrix, $\Sigma$ is an $n \times m$ rectangular diagonal matrix with non-negative real numbers on the diagonal, $V$ is an $m \times m$ unitary matrix, and $V^*$ is the conjugate transpose of $V$; see [34]. Then from (4), we obtain

$$A = YX^\dagger = YV\Sigma^{-1}U^*, \tag{5}$$

where $X^\dagger$ is the pseudoinverse of $X$; see [35]. It should be noted that calculating the eigendecomposition of the $n \times n$ matrix $A$ can be prohibitively expensive if $n$ is large, i.e., if $n \gg m$. As a result, the goal is to compute eigenvectors and eigenvalues without explicitly representing or manipulating $A$. A low-rank approximation matrix $\tilde{A}$ is constructed for this purpose, and its eigendecomposition is calculated to obtain the DMD modes and eigenvalues. The *DMD modes* and *DMD eigenvalues* are intended to approximate the eigenvectors and eigenvalues of $A$.

A reduced SVD of $X = U_r\Sigma_r V_r^*$ can be used to obtain the low-rank approximation matrix $\tilde{A}$, where $U_r$ is $n \times r$, $\Sigma_r$ is $r \times r$ diagonal, $V_r$ is $m \times r$, and $r$ is the rank of $X$, $(r \leq m)$. Then, using (5), we obtain the low-dimensional representation

$$\tilde{A} = U_r^* A U_r = U_r^* Y V_r \Sigma_r^{-1}. \tag{6}$$

The following algorithm (Algorithm 1) provides a robust method for computing DMD modes and eigenvalues.

In its original form [1], the algorithm of the DMD method differs slightly from the one described above. The only difference is that the DMD modes (at step 5) are computed by the formula

$$\Phi = UW, \tag{7}$$

where $W$ is the eigenvector matrix of $\tilde{A}$. The DMD modes calculated by Algorithm 1 are called *exact DMD modes*, because Tu et al. in [16] prove that these are exact eigenvectors of matrix $A$. The modes computed by (7) are referred to as *projected DMD modes*. It is worth noting that the DMD method is generalized and extended to a larger class of data sets in [16], where the assumption of evenly spaced measurements was relaxed.

---

**Algorithm 1** Exact DMD

---

**Input**: Data matrices  $X$  and $Y$, and rank $r$.
**Output**: DMD modes  $\Phi$   and eigenvalues $\Lambda$

1: **Procedure** DMD(X,Y,r).
2:    $[U, \Sigma, V] = SVD(X, r)$      *(Reduced r-rank SVD of X)*
3:    $\tilde{A} = U^*YV\Sigma^{-1}$        *(Low-rank approximation of A)*
4:    $[W, \Lambda] = EIG(\tilde{A})$        *(Eigen-decomposition of Ã)*
5:    $\Phi = YV\Sigma^{-1}W$        *(DMD modes of A)*
6: **End Procedure**

---

Finally, knowing the DMD modes $\Phi$ and eigenvalues $\Lambda = diag\{\lambda_i\}$, we can reconstruct the time series of data set $Z$ in (1) by expression

$$\hat{\mathbf{x}}_k = \Phi\Lambda^k\mathbf{b}\,, \tag{8}$$

where $b = \Phi^\dagger x_0$ is the vector of the initial amplitudes of DMD modes.

The DMD discrete-time eigenvalues $\lambda_j$ can also be converted to continuous time eigenvalues (Fourier modes)

$$\omega_j = \frac{\ln(\lambda_j)}{\triangle t}\,, \ j = 1,\ldots,r.$$

A continuous time dynamical system can be reconstructed as a function of time by the expression

$$\hat{\mathbf{x}}(t) = \Phi\exp(\Omega t)\mathbf{b}\,, \tag{9}$$

where $\Omega = diag\{\omega_1,\ldots,\omega_r\}$. A prediction of the future state of the system is obtained from the expression (9) for any time $t$.

*1.2. Matrix Similarity*

Here, we will briefly describe an important matrix technique called *similarity transformation*, which we will use in the next section.

**Definition 1.** *Let A and B be $n \times n$ matrices. If there is a non-singular $n \times n$ matrix P exists such that*

$$A = P^{-1}BP$$

*then we say that A and B are similar to each other.*

We will state some well-known properties of similar matrices; see [36].

**Lemma 1.** *If A and B are similar, then they have the same rank.*

**Lemma 2.** *If A and B are similar, then they have the same eigenvalues.*

It is easy to show that if $A$ and $B$ are similar and $\mathbf{x}$ is an eigenvector of $B$, then $P^{-1}\mathbf{x}$ is an eigenvector of $A = P^{-1}BP$.

**2. New DMD Algorithms**

In this section, we introduce two new alternatives to the standard DMD algorithm.

*2.1. An Alternative of Exact DMD Algorithm*

The DMD algorithms presented in the previous section use the advantage of low dimensionality in the data to make a low-rank approximation of the operator $A$ that best approximates the nonlinear dynamics of the data set.



We suggest that the modal structures can be extracted from the following matrix

$$\hat{A} = \Sigma_r^{-1} U_r^* Y V_r, \tag{10}$$

rather than the matrix $\tilde{A}$ defined by (6). The two matrices $\tilde{A}$ and $\hat{A}$ are similar, with transformation matrix $\Sigma_r$

$$\hat{A} = \Sigma_r^{-1} \tilde{A} \Sigma_r, \tag{11}$$

they therefore have the same eigenvalues. As a result, the eigenvectors of the matrix $\tilde{A}$ can be expressed in terms of the eigenvectors of $\hat{A}$.

Let

$$\hat{A}\hat{W} = \hat{W}\Lambda \tag{12}$$

be eigendecomposition of matrix $\hat{A}$. Then, using relations (7), (11) and (12) we can easily obtain the following expression:

$$\tilde{A}\Sigma_r \hat{W} = \Sigma_r \hat{W}\Lambda,$$

which yields the formula

$$\Phi = U_r \Sigma_r \hat{W}, \tag{13}$$

for the DMD modes. Expression (13) corresponds to projected DMD modes defined by (7). To be thorough, we will prove that matrix

$$\hat{\Phi} = Y V_r \hat{W}, \tag{14}$$

corresponds to exact DMD modes $\Phi = YVW\Sigma^{-1}$ defined at Step 5 in Algorithm 1; see [37].

**Theorem 1.** *Let $(\lambda, \mathbf{w})$, with $\lambda \neq 0$, be an eigenpair of $\hat{A}$ defined by (10), then the corresponding eigenpair of $A$ is $(\lambda, \varphi)$, where*

$$\varphi = YV_r \mathbf{w}.$$

**Proof.** By using reduced SVD $X = U_r \Sigma_r V_r^*$ and the pseudoinverse of $X$

$$X^\dagger = V_r \Sigma_r^{-1} U_r^*,$$

we obtain the expression

$$A = YX^\dagger = YV_r \Sigma_r^{-1} U_r^*.$$

Let us now express $A\varphi$

$$A\varphi = YV_r \Sigma^{-1} U_r^* YV_r \mathbf{w}$$

which implies, by using (10)

$$A\varphi = YV_r \hat{A}\mathbf{w} = YV_r \mathbf{w}\lambda = \lambda\varphi.$$

In addition, $\varphi \neq 0$, since if $YV_r \mathbf{w} = 0$, then $\Sigma_r^{-1} U_r^* YV_r \mathbf{w} = \hat{A}\mathbf{w} = 0$. This implies that $\lambda = 0$. Hence, $\varphi$ is an eigenvector of $A$ with eigenvalue $\lambda$. The proof is completed. □

Now, we are ready to formulate an alternative to the exact DMD method described (Algorithm 1); see Algorithm 2 below.

According to Theorem 1, modes in (14) generated by Algorithm 2 are eigenvectors of the matrix $A$. Although the matrices $\hat{A}$ and $\tilde{A}$ are computationally similar because they have the same but permuted multipliers, the calculation of DMD modes $\hat{\Phi}$ in Algorithm 2 is more economical than the calculation of modes $\Phi$ in Algorithm 1.

---

**Algorithm 2** Alternative exact DMD

---
**Input**: Data matrices $X$ and $Y$, and rank $r$.
**Output**: DMD modes $\hat{\Phi}$ and eigenvalues $\Lambda$

1: **Procedure** DMD(X,Y,r).
2:     $[U, \Sigma, V] = SVD(X, r)$     *(Reduced r-rank SVD of X)*
3:     $\hat{A} = \Sigma^{-1}U^*YV$     *(Low-rank approximation of A)*
4:     $[\hat{W}, \Lambda] = EIG(\tilde{A})$     *(Eigen-decomposition of $\hat{A}$)*
5:     $\hat{\Phi} = YV\hat{W}$     *(DMD modes of A)*
6: **End Procedure**

---

### 2.2. A New DMD Algorithm for Full Rank Dataset

We will assume in this section that matrix $X \in R^{n \times m}$ is a full rank matrix, i.e., $r = m$, where $n > m$ and $r = \text{rank}(X)$. Our goal is to obtain a more efficient algorithm for calculating DMD modes and eigenvalues in this particular case.

We suggest that the modal structures can be extracted from the following matrix

$$\bar{A} = V_r \hat{A} V_r^*, \tag{15}$$

where $V_r$ is the unitary matrix from the SVD of $X = U_r \Sigma_r V_r^*$. Matrices $\hat{A}$ and $\bar{A}$ are obviously similar. From (10) and (15), we obtain the expression

$$\bar{A} = V_r \Sigma_r^{-1} U_r^* Y, \tag{16}$$

which yields

$$\bar{A} = X^\dagger Y, \tag{17}$$

where $X^\dagger$ is the Moore–Penrose pseudoinverse of $X$. Denoting the eigen-decomposition of $\bar{A}$ by

$$\bar{A}\bar{W} = \bar{W}\Lambda, \tag{18}$$

where the columns of $\bar{W}$ are eigenvectors and $\Lambda$ is a diagonal matrix containing the corresponding eigenvalues. From Definition of $\hat{A}$, and relations (15) and (18), we deduce

$$AU_r \Sigma_r V_r^* \bar{W} = U_r \Sigma_r V_r^* \bar{W}\Lambda \tag{19}$$

or equivalently

$$A(X\bar{W}) = (X\bar{W})\Lambda. \tag{20}$$

Thus, we show that

$$\Phi = X\bar{W} \tag{21}$$

is the matrix of DMD modes. These are the projected DMD modes (see Theorem 3 below). We will express the exact DMD modes in the next Theorem.

**Theorem 2.** *Let $(\lambda, \bar{w})$, with $\lambda \neq 0$, be an eigenpair of $\bar{A}$ defined by (17). Then, the corresponding eigenpair of A is $(\lambda, \bar{\varphi})$, where*

$$\bar{\varphi} = Y\bar{w}. \tag{22}$$

**Proof.** Let us express $A\varphi$ by using (4)

$$A\bar{\varphi} = YV_r \Sigma_r^{-1} U_r^* Y\bar{w}.$$

From the last relation and (16), we get

$$A\bar{\varphi} = Y\bar{A}\bar{w} = \lambda Y\bar{w} = \lambda\bar{\varphi}.$$

Furthermore, $\bar{\varphi} \neq 0$, because if $Y\bar{w} = 0$, then $V_r \Sigma_r^{-1} U_r^* Y\bar{w} = \bar{A}\bar{w} = 0$, implying $\lambda = 0$. Hence, $\bar{\varphi}$ is an eigenvector of $A$ with an eigenvalue $\lambda$. $\quad\square$

Next, we resume the results from above in the form of an algorithm.

We intentionally omitted Step 1 of Algorithm 1 (or Algorithm 2) in Algorithm 3, because the pseudo-inverse matrix of full-rank matrix $X$ can be calculated not only by SVD but also by formula

$$X^\dagger = (A^*A)^{-1}A^*.$$

The presented Algorithm 3 has the greatest advantage among the described algorithms from a computational point of view in the case of full-rank data. We will now prove that modes in expression (21) are projected DMD modes.

---

**Algorithm 3** DMD Algorithm for full rank dataset

---

**Input**: Data matrices $X$ and $Y$.
**Output**: DMD modes $\bar{\Phi}$ and eigenvalues $\Lambda$

1: **Procedure** DMD(X,Y).
2: $\quad \bar{A} = X^\dagger Y \qquad$ *(Low-rank approximation of A)*
3: $\quad [\bar{W}, \Lambda] = EIG(\bar{A})$ *(Eigen-decomposition of $\bar{A}$)*
4: $\quad \bar{\Phi} = Y\bar{W}$ *(DMD modes of A)*
5: **End Procedure**

---

**Theorem 3.** *Let $(\lambda, \bar{\mathbf{w}})$, with $\lambda \neq 0$, be an eigenpair of $\bar{A}$ defined by (15), and let $P_X$ denotes the orthogonal projection matrix onto the column space of $X$. Then, the vector*

$$\varphi = X\bar{\mathbf{w}} \tag{23}$$

*is an eigenvector of $P_X A$ with an eigenvalue $\lambda$. Furthermore, if $\bar{\varphi} = Y\bar{\mathbf{w}}$ is given by (22), then $P_X \bar{\varphi} = \lambda\varphi$.*

**Proof.** From the reduced SVD $X = U_r \Sigma_r V_r^*$, we obtain the orthogonal projection onto the column space of $X$ by $P_X = XX^\dagger = U_r U_r^*$. From (17) and the relation $Y = AX$, we get

$$X^\dagger AX = \bar{A},$$

which implies

$$P_X A\varphi = XX^\dagger AX\bar{\mathbf{w}} = X\bar{A}\bar{\mathbf{w}} = \lambda X\bar{\mathbf{w}} = \lambda\varphi.$$

According to the previous expression, $\varphi$ is an eigenvector of $P_X A$ with an eigenvalue $\lambda$. Let us now express

$$P_X \bar{\varphi} = XX^\dagger Y\bar{\mathbf{w}} = U_r \Sigma_r (\Sigma_r^{-1} U_r^* Y V_r)\bar{\mathbf{w}} = X\bar{A}\bar{\mathbf{w}} = \lambda X\bar{\mathbf{w}} = \lambda\varphi.$$

which proves the statement of the Theorem. $\quad\square$

*2.3. In Terms of Companion Matrix*

Let us consider the case where the last snapshot $x_m$ in the data set (1) is in the column space of $X$, i.e. $\mathbf{x}_m$ is a linear combination of $\mathbf{x}_0, \ldots, \mathbf{x}_{m-1}$. Therefore

$$\mathrm{Im}(Y) \subset \mathrm{Im}(X).$$

In this case, matrix $\bar{A}$ defined by (16) is in type of the Frobenius companion matrix and it relates the data sets exactly $Y = X\bar{A}$, even if the data are generated by nonlinear dynamics. Moreover, in this case, the projected DMD modes (23) and exact DMD modes (22) are identical.

**Theorem 4.** *If the columns of Y are spanned by those of X the DMD modes (23) are eigenvectors of $\bar{A}$ defined by (17).*

**Proof.** From the statement of the Theorem, it follows

$$P_X Y = Y,$$

where $P_X = XX^\dagger = U_r U_r^*$ is the orthogonal projection onto the image of $X$. We obtain from the previous relation and the reduced SVD of $X$

$$P_X A = P_X Y V_r \Sigma_r^{-1} U_r^* = Y V_r \Sigma_r^{-1} U_r^* = A.$$

Finally, we can show that $\varphi = X\bar{\mathbf{w}}$ defined by (23) is an eigenvector of $A$. The following relations are fulfilled

$$A\varphi = AX\bar{\mathbf{w}} = P_X AX\bar{\mathbf{w}} = XX^\dagger AX\bar{\mathbf{w}} = X\bar{A}\bar{\mathbf{w}} = \lambda X\bar{\mathbf{w}} = \lambda\varphi,$$

which proves the Theorem. □

In this case, a reconstruction of the data matrix $Y$, using relation $Y = X\bar{A}$ yields

$$Y = X\bar{W}\Lambda\bar{W}^{-1},$$

where $\bar{A} = \bar{W}\Lambda\bar{W}^{-1}$ is the eigendecomposition of $\bar{A}$ defined by (18). Using that $\bar{A}$ is a Frobenius companion matrix, and notation (21), we obtain

$$Y = \Phi\Lambda V(\lambda) \tag{24}$$

where $V(\lambda)$ is a Vandermonde matrix, i.e.

$$Y = \begin{pmatrix} | & | & \\ \phi_1 & \phi_2 & \dots \\ | & | & \end{pmatrix} \begin{pmatrix} \lambda_1 & 0 & \dots \\ 0 & \lambda_2 & \dots \\ \vdots & \vdots & \ddots \end{pmatrix} \begin{pmatrix} 1 & \lambda_1 & \dots & \lambda_1^{m-2} \\ 1 & \lambda_2 & \dots & \lambda_2^{m-2} \\ \vdots & \vdots & \dots & \vdots \end{pmatrix}. \tag{25}$$

where $\phi_i$ and $\lambda_i$ are DMD modes and eigenvalues, respectively. In this formulation, each mode $\phi_i$ is scaled with associated $\lambda_i$. The Vandermonde matrix captures the iterative exponentiation of the DMD eigenvalues. The representation (24) and (25) gives us a factorization of the data into spatial modes, amplitudes, and temporal dynamics. Moreover, the amplitudes in this case coincide with the DMD eigenvalues and do not depend on the initial condition.

*2.4. Computational Cost and Memory Requirement*

Table 1 gives a brief summary of the main matrices in the three algorithms considered. The representations of the corresponding reduced order approximations of the Koopman operator are shown, as well as the formulas for calculating the DMD modes in three cases.

**Table 1.** Reduced matrices and DMD modes.

| | Algorithm 1 $(r \leq m)$ | Algorithm 2 $(r < m)$ | Algorithm 3 $(r = m)$ |
|---|---|---|---|
| *Reduced matrix* | $\tilde{A} = U_r^* Y V_r \Sigma_r^{-1}$ | $\hat{A} = \Sigma_r^{-1} U_r^* Y V_r$ | $\bar{A} = X^\dagger Y$ |
| *DMD modes* | $\Phi = Y V_r \Sigma_r^{-1} W$ | $\hat{\Phi} = Y V_r \hat{W}$ | $\bar{\Phi} = Y\bar{W}$ |

Although the structures of the three low-rank approximation matrices $\tilde{A}$, $\hat{A}$ and $\bar{A}$ are similar, the corresponding representations $\hat{\Phi}$ and $\bar{\Phi}$ have a simpler form when determining the DMD modes than $\Phi$. In Algorithm 2, three matrices need to be stored and two matrix

multiplications performed, while in Algorithm 3, it is necessary to store only two matrices and perform one matrix multiplication.

Since the reduced matrix $\tilde{A}$ in Algorithm 1 is of the same size as the corresponding matrices $\hat{A}$ and $\bar{A}$ in the alternative algorithms, they therefore require the same resources to compute their spectral decompositions. To estimate the computational cost for the three algorithms considered, we will ignore the comparable computations and focus on the different ones. While in Algorithms 1 and 2, the calculation of the corresponding reduced matrices $\tilde{A}$ and $\hat{A}$ involves SVD of $X$ and matrix multiplication, in Algorithm 3, matrix $\bar{A}$ is calculated by the pseudo-inverse matrix of $X$. The DMD modes for the three algorithms are calculated by the corresponding matrix multiplications, as indicated in Table 1. The computational costs are shown in Table 2, see Golub and Van Loan [38].

**Table 2.** Computational costs.

| Cost of | Algorithm 1 | Algorithm 2 | Algorithm 3 |
|---|---|---|---|
| *SVD of* $X$ | $6nm^2 + 20m^3$ | $6nm^2 + 20m^3$ | $-$ |
| *Reduced matrix* | $2r^3 + 2nr^2 + r$ | $2r^3 + 2nr^2 + r$ | $n^2r + nr^2$ |
| *DMD modes* $\Phi$ | $2r^3 + 2nr^2 + r$ | $2r^3 + (2n-2)r^2$ | $nr^2$ |
| *Total cost* | $6nm^2 + 20m^3 + 4r^3 + 4nr^2$ | $6nm^2 + 20m^3 + 4r^3 + (4n-2)r^2$ | $n^2r + 2nr^2$ |

From the memory point of view, the corresponding matrices that require the same amount of memory for all three algorithms are: the data matrix $Y$, the reduced matrix ($\tilde{A}, \hat{A}$ or $\bar{A}$), and the eigenvectors matrix ($W, \hat{W}$ or $\bar{W}$). The number of floating point numbers to be stored for the corresponding reduced-order matrix and eigenvector matrix is equal to $r^2$ in all three algorithms. The difference in the required memory for the three algorithms is determined by the matrices needed to calculate the DMD modes. The number of floating point numbers that must be stored for the DMD calculations is shown in Table 3.

**Table 3.** Memory requirements for DMD mode matrices.

| Matrix | Algorithm 1 | Algorithm 2 | Algorithm 3 |
|---|---|---|---|
| $Y$ | $nm$ | $nm$ | $nm$ |
| $V_r$ | $rm$ | $rm$ | $-$ |
| $\Sigma_r^{-1}$ | $r$ | $-$ | $-$ |
| Total memory | $(n+r)m + r$ | $(n+r)m$ | $nm$ |

## 3. Numerical Illustrative Examples

In this section, we will compare the results obtained by the standard DMD algorithm and the new algorithms (Algorithms 2 and 3) introduced in Section 2. All considered examples are well known in the literature. All numerical experiments and simulations were performed on Windows 7 with MATLAB release R2013a on Acer Aspire 571G laptop with an Intel(R) Core(TM) i3-2328M CPU @2.2GHz processor and 4 GB RAM.

We should note that all three algorithms that we consider in the present work require the use of some of the most expensive functions from a computational point of view: *svd* and *eig*, respectively, for calculating SVD and spectral decomposition of matrices.

### 3.1. Example 1: Spatiotemporal Dynamics of Two Signals

We consider an illustrative example of two mixed spatiotemporal signals

$$f_1(x,t) = sech(x+6)e^{i3.8t} \text{ and } f_2(x,t) = 2sech(x)tanh(x)e^{i2.2t}$$

and the mixed signal

$$X(t) = f_1(x,t) + f_2(x,t). \tag{26}$$

The two signals $f_1$, $f_2$ and mixed signal $X$ are illustrated in Figure 1a–c.

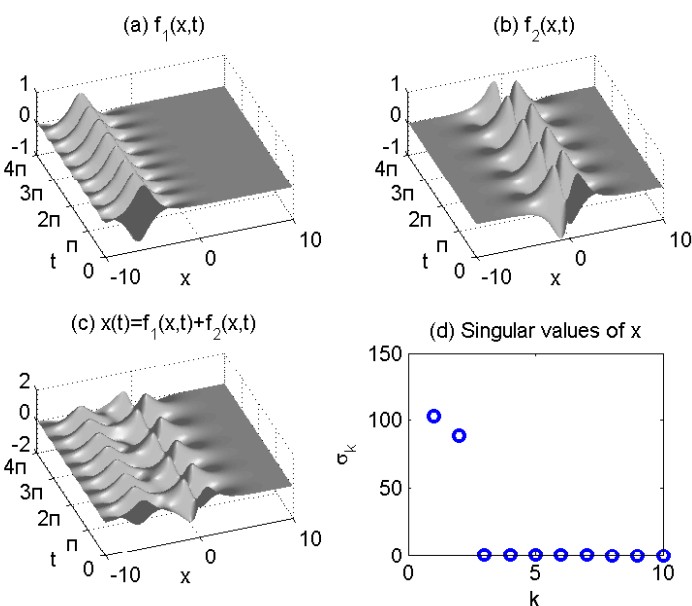

**Figure 1.** Spatiotemporal dynamics of two signals (**a**) $f_1(x,t)$, (**b**) $f_2(x,t)$, and mixed signal in (**c**) $x = f_1 + f_2$. Singular values of $X$ are shown in (**d**).

Figure 1d depicts the singular values of data matrix $X$, indicating that the data can be adequately represented by the rank $r = 2$ approximation.

We perform a rank-2 DMD reconstruction of data by using standard DMD (Algorithm 1) and Alternative DMD (Algorithm 2). These reconstructions are shown in Figure 2a,b.

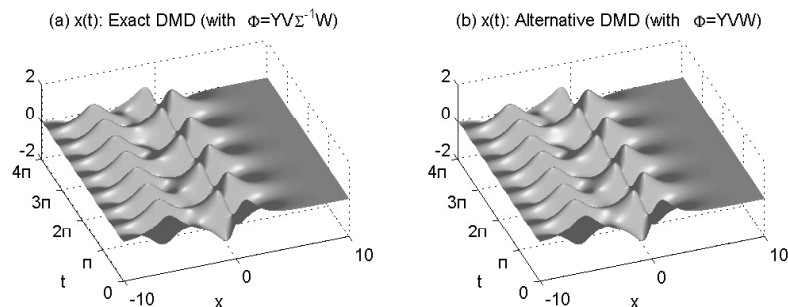

**Figure 2.** Rank$-2$ reconstructions of the signal $X$ by: standard DMD (**a**) and Alternative DMD (**b**).

The two reconstructions are nearly exact approximations, with the DMD modes and eigenvalues matching those of the underlying signals $f_1$ and $f_2$ perfectly. Both algorithms reproduce the same continuous-time DMD eigenvalues $\omega_1 = 2.2i$ and $\omega_2 = 3.8i$. Their imaginary components correspond to the frequencies of oscillation.

Figure 3 panels compare the first two DMD modes, with true modes plotted alongside modes extracted by *Standard DMD* (Algorithm 1) and *Alternative DMD* (Algorithm 2). The DMD modes produced by the two algorithms match exactly to nearly machine precision.

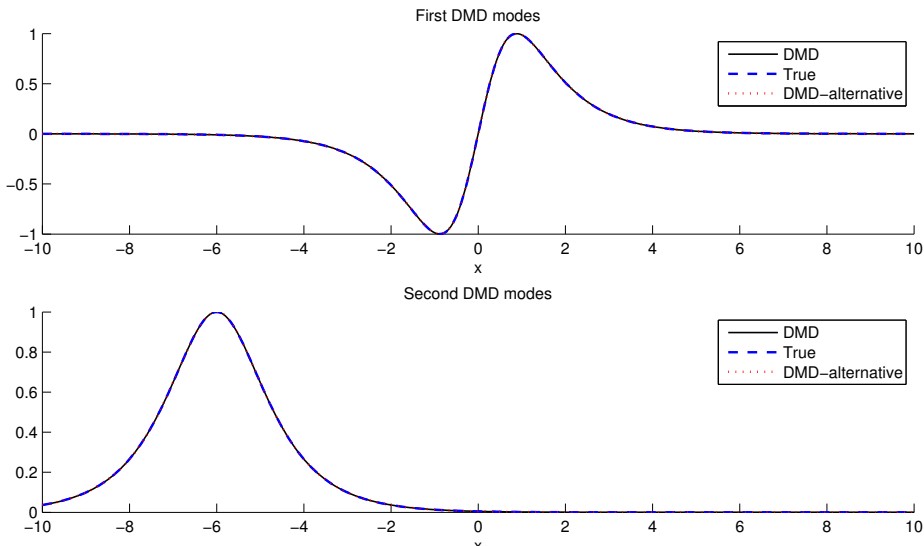

**Figure 3.** Firts two DMD modes: true modes, modes extracted by standard DMD and modes extracted by Alternative DMD.

Table 4 compares the execution time results of simulations using Algorithms 1 and 2.

**Table 4.** Execution time (in sec.) by Algorithms 1 and 2.

| Number of Cycles (k) | *Standard DMD* (Algorithm 1) | *Alternative DMD* (Algorithm 2) |
|:---:|:---:|:---:|
| $k = 1000$ | 0.5407 | 0.4864 |
| $k = 10000$ | 4.2264 | 4.2013 |

*3.2. Example 2: Re = 100 Flow around a Cylinder Wake*

We consider a time series of fluid vortex fields for the wake behind a round cylinder at a Reynolds number Re = 100. The Reynolds number is defined as $Re = DU_\infty/v$, where $D$ is the cylinder diameter, $U_\infty$ is the free-stream velocity, and $v$ is the kinematic fluid viscosity. It quantifies the ratio of inertial to viscous forces.

This example is taken from [17], see also [39]. We use the same data set which is publicly available at 'www.siam.org/books/ot149/flowdata'. Collected data consists $m = 150$ snapshots at regular intervals in time, $10\triangle t$ , sampling five periods of vortex shedding. An example of a vorticity field is shown in Figure 4.

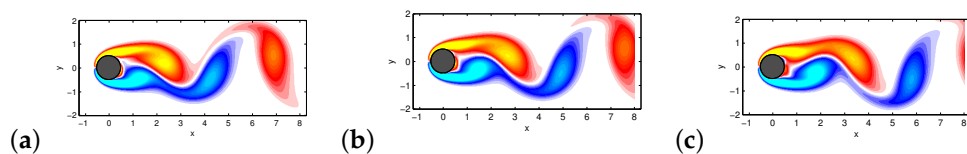

(**a**)　　　　　　　　(**b**)　　　　　　　　(**c**)

**Figure 4.** Some vorticity field snapshots for the wake behind a cylinder at $Re = 100$ are shown in (**a**–**c**).

We performed Algorithms 1 and 2 to obtain DMD decomposition and reconstruction of the data. Two algorithms reproduce the same DMD eigenvalues and modes.

The quality of low-rank approximations is measured by the relative error $e_{DMD}$

$$e_{DMD} = \frac{\|\mathbf{x} - \hat{\mathbf{x}}\|_2}{\|\mathbf{x}\|_2}, \tag{27}$$

where $\hat{x}$ is DMD reconstruction of data by using expression (9). Actually, both standard DMD and alternative DMD reconstructions have the same error, see Table 5.

**Table 5.** Relative errors for DMD reconstructions by Algorithms 1 and 2.

|  | *Standard DMD* | *Alternative DMD* |
|---|---|---|
| Relative errors | $e_{st.DMD} = 5.3685 \times 10^{-4}$ | $e_{alt.DMD} = 5.3685 \times 10^{-4}$ |

See Figure 5 for DMD eigenvalues and singular values of the data matrix $X$.

*(a) First 20 singular values of X.*      *(b) 'o':by Algorithm 1; '+':by Algorithm 2.*

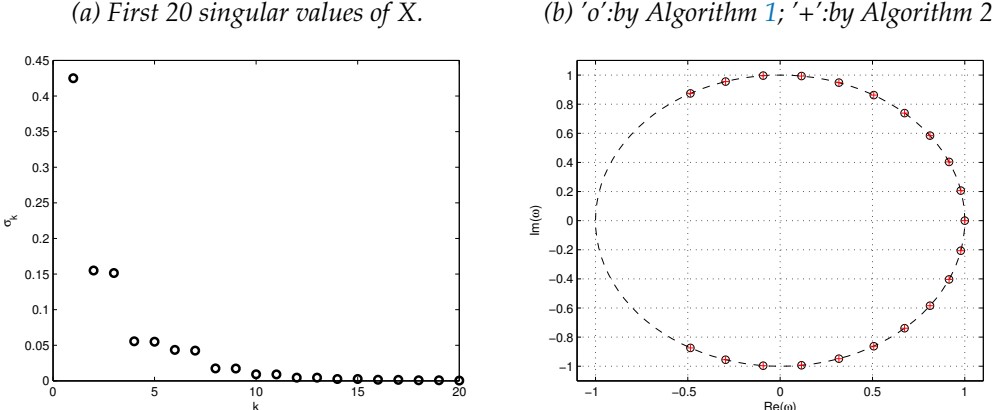

**Figure 5.** Singular values of $X$ (**a**) and DMD eigenvalues computed by Algorithms 1 and 2 (**b**).

Figure 6 shows the first six DMD modes computed by Algorithms 1 and 2, respectively. The only difference is in the visualization, with contour lines added to the DMD modes obtained by Algorithm 2 (otherwise we obtain the same picture).

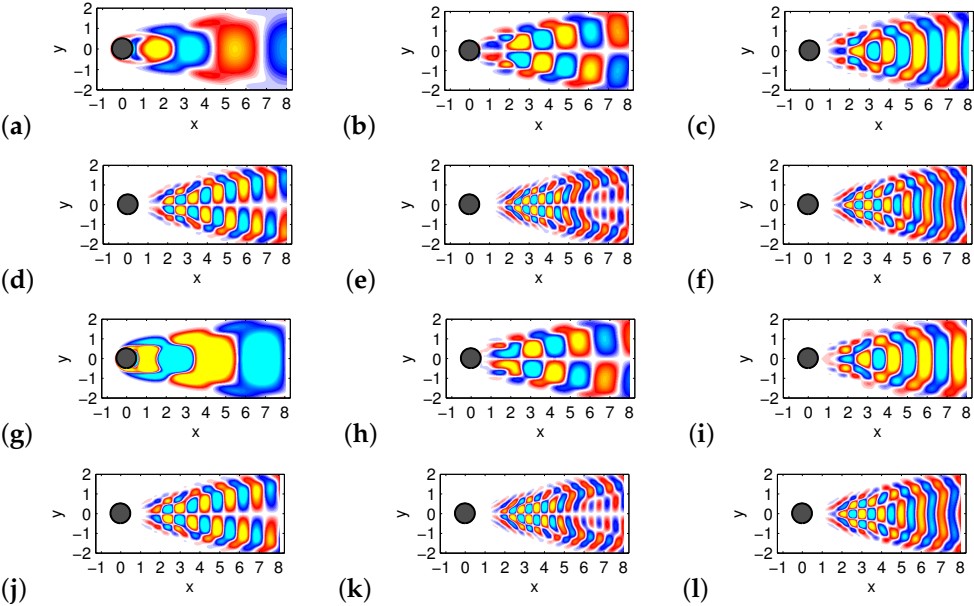

**Figure 6.** The first six DMD modes computed by Algorithm 1 are shown in (**a**–**f**). Corresponding DMD modes computed by Algorithm 2 are in (**g**–**l**).

It can be seen from Figures 5 and 6 that the two algorithms produce the same DMD eigenvalues and DMD modes.

Table 6 shows the execution time of this task by Algorithms 1 and 2.

**Table 6.** Execution time (in sec.) by Algorithms 1 and 2.

| Number of Cycles (k) | Standard DMD (Algorithm 1) | Alternative DMD (Algorithm 2) |
|---|---|---|
| $k = 100$ | 15.6083 | 15.5615 |
| $k = 1000$ | 156.3974 | 154.8962 |

*3.3. Example 3: DMD with Different Koopman Observables*

We consider the nonlinear Schrödinger (NLS) equation

$$i\frac{\partial q}{\partial t} + \frac{1}{2}\frac{\partial^2 q}{\partial \xi^2} + |q|^2 q = 0, \tag{28}$$

where $q(\xi, t)$ is a function of space and time. This equation can be rewritten in the equivalent form

$$\frac{\partial q}{\partial t} = \frac{i}{2}\frac{\partial^2 q}{\partial \xi^2} + i|q|^2 q = 0. \tag{29}$$

Fourier transforming in $\xi$ gives the differential equation in the Fourier domain variables $\hat{q}$

$$\frac{\partial \hat{q}}{\partial t} = \frac{ik^2}{2}\hat{q} + i\widehat{|q|^2 q} = 0. \tag{30}$$

By discretizing in the spatial variable we can generate a numerical approximation to the solution of (7); see [17].

The following parameters were used in the simulation: there is a total of 21 slices of time data over the interval $t \in [0, 2\pi)$, the state variables are an $n$-dimensional discretization of $q(\xi, t)$, so that $q(\xi, t_k) \to \xi_k$, where $n = 400$. The resulting $\xi_k$ are the columns of the generated data matrix. We analyze the two-soliton solution that has the initial condition $q(\xi, 0) = 2\text{sech}(\xi)$. The result of this simulation is shown in Figure 7a.

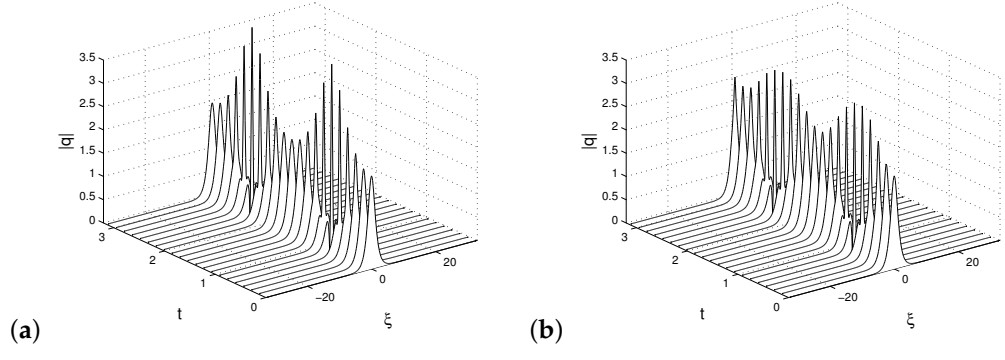

(a)                                             (b)

**Figure 7.** The full simulation of the NLS Equation (7) in (**a**) and DMD reconstruction (**b**) by standard DMD algorithm, where the observable is given by $g_{DMD}(\mathbf{x}) = \mathbf{x}$, where $\mathbf{x} = q(\xi, t)$, in panel (**b**).

We performed low-rank DMD approximation (r = 10) with standard DMD method as shown in Figure 7b. In this case, by standard DMD approximation, it is meant that the state vectors $\mathbf{x}$ coincide with the Koopman quantities

$$g_{DMD}(\mathbf{x}) = \mathbf{x} = q(\xi, t).$$

The obtained approximation is not satisfactory.

To improve the approximation, we can use another Koopman observable

$$g_1(\mathbf{x}) = \begin{pmatrix} \mathbf{x} \\ |\mathbf{x}|^2 \mathbf{x} \end{pmatrix}, \tag{31}$$

which is based on the NLS nonlinearity; see also [17].

In this case, we define new input data matrices corresponding to $X$ and $Y$ defined by (3), as follows

$$X1 = \begin{pmatrix} X \\ |X|^2X \end{pmatrix} \text{ and } Y1 = \begin{pmatrix} Y \\ |Y|^2Y \end{pmatrix},$$

respectively. Following that, the DMD approach is used in the usual way with matrices $X1$ and $Y1$ instead of $X$ and $Y$. New approximation gives a superior reconstruction, which is evident from Figure 8. We have performed DMD reconstructions using both algorithms Algorithms 1 and 2.

It can be seen from Figure 9b that both algorithms reproduce the same DMD eigenvalues. To measure the quality of approximations, the relative error formula defined by (27) is used. Both reconstructions, by standard DMD (Algorithm 1) and alternative DMD (Algorithm 2), have the same error curve; see Figure 9a.

Algorithms 1 and 2 are compared in terms of execution times, the results are included in Table 7.

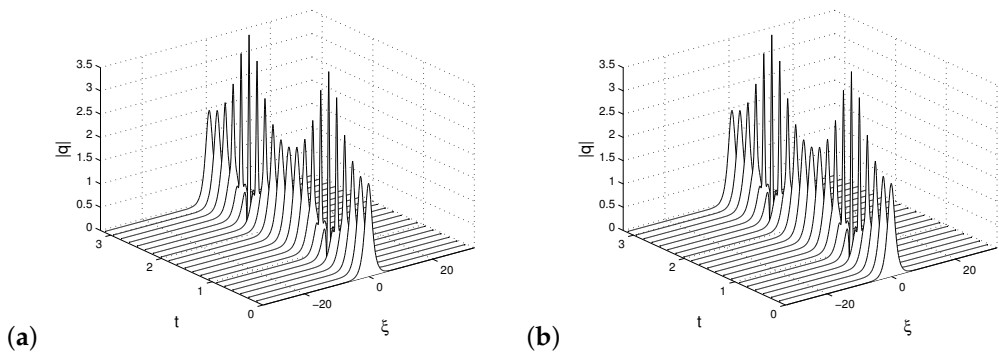

(**a**)                        (**b**)

**Figure 8.** DMD reconstructions, based on new observable $g_1$ defined by (31). Reconstruction by Algorithm 1 in (**a**) and by Algorithm 2 in (**b**).

(**a**) Error curves by Algorithms 1 and 2.      (**b**) 'o':by Algorithm 1; '+':by Algorithm 2.

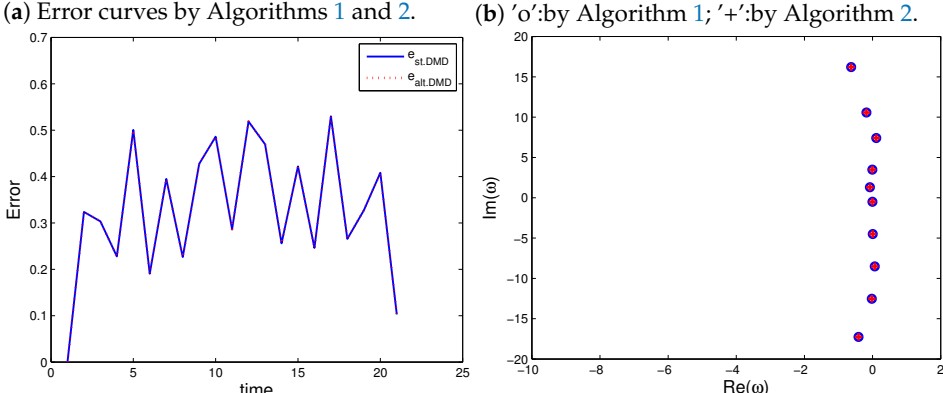

**Figure 9.** Relative errors (**a**) and DMD eigenvalues (**b**).

**Table 7.** Execution time (in sec.) by Algorithms 1 and 2.

| Number of Cycles (k) | *Standard DMD* (Algorithm 1) | *Alternative DMD* (Algorithm 2) |
|---|---|---|
| $k = 1000$ | 0.5873 | 0.5295 |
| $k = 10000$ | 5.5346 | 5.0998 |

### 3.4. Example 4: Standing Wave

It is known that the standard DMD algorithm is not able to represent a standing wave in the data [16]. For example, if measurements of a single sine wave are collected, DMD fails to capture periodic oscillations in the data.

In this case, the data matrix $X$ contains a single row

$$X = \left[ \begin{array}{cccc} x_1 & x_2 & \ldots & x_m \end{array} \right], \tag{32}$$

where each $x_i$ is a scalar and DMD fails to reconstruct the data. There is a simple solution to this rank deficiency problem, which involves stacking multiple time-shifted copies of the data into augmented data matrices

$$X_{aug} = \left[ \begin{array}{cccc} x_1 & x_2 & \ldots & x_{m-s} \\ x_2 & x_3 & \ldots & \ldots \\ \vdots & \vdots & \ddots & \vdots \\ x_s & x_{s+1} & \ldots & x_{m-1} \end{array} \right]; \; Y_{aug} = \left[ \begin{array}{cccc} x_2 & x_3 & \ldots & x_{m-s+1} \\ x_3 & x_4 & \ldots & \ldots \\ \vdots & \vdots & \ddots & \vdots \\ x_{s+1} & x_{s+2} & \ldots & x_m \end{array} \right]. \tag{33}$$

Thus, using delay coordinates achieves an increase in the rank of the data matrix $X_{aug}$. In fact, we can increase the number $s$ of delay coordinates until the data matrix reaches full rank numerically. Then, we perform the DMD technique on the augmented matrices $X_{aug}$ and $Y_{aug}$.

We can demonstrate the rank mismatch issue with an example from finance by considering the evolution in the price of only one type of commodity. In fact, this problem is quite similar to the standing wave problem. Let us consider the price evolution of the Brent Crude Oil for the period 1 February 2022–28 February 2022, containing 20 trading days; see Figure 10.

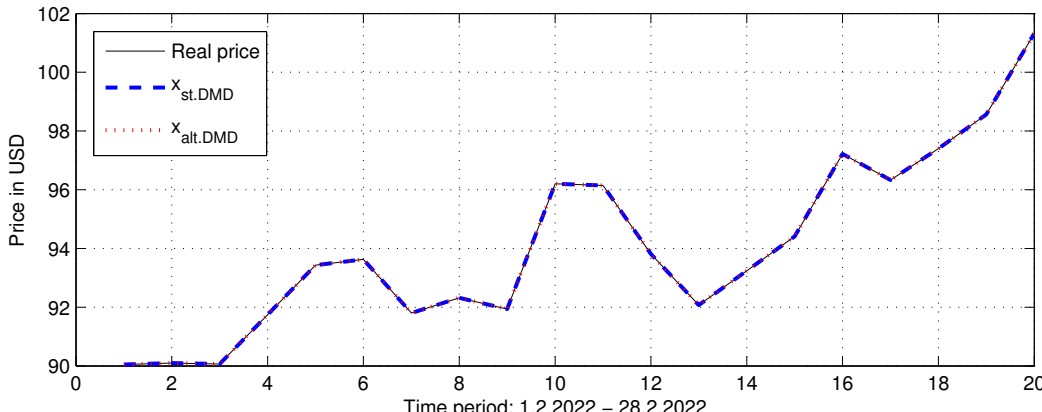

**Figure 10.** Two approximations of Brent Crude Oil price for the period 1 February 2022–28 February 2022 by Standard DMD and Alternative DMD approaches.

The data matrix $X$ is a single row as in (32) containing $m = 20$ elements, where each $x_i$ is the closing price on the corresponding day. We construct the augmented matrices $X_{aug}$ and $Y_{aug}$ as in (33). In this case, we can choose $s \in [10, 18]$, which ensures that matrices $X_{aug}$ and $Y_{aug}$ will have more rows than columns. For each $s$ in this interval, we obtain a full rank matrix $X_{aug}$.

Therefore, in this case, we can use the alternative DMD algorithm for full rank data matrices, Algorithm 3. We performed Algorithms 1 and 3 on augmented data matrices $X_{aug}$ and $Y_{aug}$ for each $s \in [10, 18]$. The results show that the best approximation of the measured data is obtained at the highest rank of $X_{aug}$, $r = 10$, with $s = 10$. In each case, the two algorithms reproduce the same approximation. Figure 10 shows the two approximations

for $\mathrm{rank}(X_{aug}) = 10$, where it can be seen that both algorithms perfectly approximate the actual price.

Execution times for Algorithms 1 and 3 are computed with the dataset of this example. Table 8 presents a comparison between the two algorithms.

**Table 8.** Execution time (in sec.) by Algorithms 1 and 3.

| Number of Cycles (k) | *Standard DMD* (Algorithm 1) | *Alternative DMD* (Algorithm 3) |
|:---:|:---:|:---:|
| $k = 1000$ | 0.1933 | 0.1324 |
| $k = 10000$ | 1.6319 | 0.9783 |

## 4. Conclusions

The purpose of this study was to introduce two new algorithms for computing approximate DMD modes and eigenvalues. We proved that each generated pairs $(\hat{\varphi}, \lambda)$ and $(\bar{\varphi}, \lambda)$ by Algorithms 2 and 3, respectively, is an eigenvector/eigenvalue pair of Koopman operator $A$ (Theorems 1 and 2). The matrices of DMD modes $\hat{\Phi}$ and $\bar{\Phi}$ from Algorithms 2 and 3 have a simpler form than the DMD mode matrix $\Phi$ from Algorithm 1. They need less memory and require fewer matrix multiplications.

We demonstrate the performance of the presented algorithms with numerical examples from different fields of application. From the obtained results, we can conclude that the introduced approaches give identical results to those of the *exact DMD method*. Comparison of simulation times shows that better effectiveness is attained by new algorithms. The presented results show that the introduced algorithms are alternatives to the standard DMD algorithm and can be used in various fields of application.

This study motivates several further investigations. Future work on the use of the proposed algorithms will consist of their application to a wider class of dynamical systems, particularly those dealing with full-rank data. Applications to other known methods that use approximate linear dynamics, such as embedding with Kalman filters, will be sought.It may be possible to develop some alternatives to some known variants of the DMD method, such as DMD with control and higher-order DMD.

An interesting direction for future work is the optimization of the introduced algorithms in relation to the required computing resources. One line of work is to implement these algorithms using parallel computing.

**Funding:** Paper written with financial support of Shumen University under Grant RD-08-144/01.03.2022.

**Conflicts of Interest:** The author declares no conflict of interest.

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
