# Peer review of "On Alternative Algorithms for Computing Dynamic Mode Decomposition"

_computation, doi:10.3390/computation10120210_

Round 1

Reviewer 1 Report

I read the manuscript with great interest. The research topic is interesting for me. Even though the manuscript has little overlap with my scientific field, reading it gave me new thoughts for solving my task, which has been tormenting me for almost 8 years now. Many thanks to the author.

1) The number of vectors x is m+1 things, maybe there is an error in formula (2), and the index k should vary from 1 to m?

2) It would be nice if the author would distinguish scalar and matrix quantities. For example, vector and matrix are bold, scalar is italic.

3) In some places it says DMD, and in some places gives the full transcript “the dynamic mode decomposition”, it would be more convenient if the author adhered to a single designation.

4) The abbreviation SVD is used, but not deciphered, the readers have to refer to [4] to get the decoding.

5) In the phrase “SVD-decomposition” the word “decomposition” occurs twice

6) There is no description of U and V, and U* (the Hermitian transpose) after equation (5)

7) It would be delightful if the author made a reference to pseudoinverse (Moore-Penrose inverse A+) matrices for non-specialists.

8) Do I understand correctly that in (6) the matrix A is substituted from (5)?

9) It would be nice if Figures 1, 2, 3, 4, 5, 6 had signed axes. Why are the abscissa axis captions in figure 1 given in the array counts?

10) Why are the eigenvalues in lines 150 and 151 given with such precision?

Reviewer 2 Report

The paper "On alternative algorithms for computing Dynamic Mode Decomposition" by Gyurhan Nedzhibov is well written and of interest for scientific community.  I recommend it for publication after addressing the following minor issues. 

In Fig.3 the curves are non-distinguishable. The author should find a way to represent it another way.  For instance, adding some panel where the discrepancies are plotted, would help to see how different the curves are.

Fig. 4 is not satisfactory. The black lines are not continuous, they are represented by a huge amount of isolated segments orthogonal to the main line (or crosses). The meaning of the colours is not described in the legend. The same for Fig. 6.

Fig.7 and 8 are drastically different in representation style from Figs.1 and 2.  I suggest to uniform these. In my view, the better option is represented in Figs.1 and 2.

Eq. (30): it is surprising that the nonlinear term in (29) (the last one) is represented in the Fourier space by a local term (the last term in (30)). Products in the physical space are usually represented as a non-local term acting on all Fourier harmonics.

line 141: misprint: 13 --> i3.  In this paragraph, I would also suggest to mention the most "expensive" Matlab functions used in this work (SVD and spectra decomposition, as well as I can see). 

To improve the paper, the author could briefly discuss how they plan to continue this line of research.  In particular, since all the computations presented in the paper were performed on a laptop, it is of interest to discuss how the presented techniques can be extended/scaled to more computationally demanding statements. (Note, in [24] a parallel method is presented).
